# Computation of Atomic Astrophysical Opacities †

**Claudio Mendoza** [1,2] 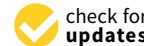

[1]   Department of Physics, Western Michigan University, Kalamazoo, MI 49008, USA;
      claudio.mendozaguardia@wmich.edu; Tel.: +1-269-370-8465

[2]   Venezuelan Institute for Scientific Research, Caracas 1020, Venezuela

†   This review is originally based on a talk given at the 12th International Colloquium on Atomic Spectra and
    Oscillator Strengths for Astrophysical and Laboratory Plasmas, São Paulo, Brasil.

**Abstract:** The revision of the standard Los Alamos opacities in the 1980–1990s by a group from the Lawrence Livermore National Laboratory (OPAL) and the Opacity Project (OP) consortium was an early example of collaborative big-data science, leading to reliable data deliverables (atomic databases, monochromatic opacities, mean opacities, and radiative accelerations) widely used since then to solve a variety of important astrophysical problems. Nowadays the precision of the OPAL and OP opacities, and even of new tables (OPLIB) by Los Alamos, is a recurrent topic in a hot debate involving stringent comparisons between theory, laboratory experiments, and solar and stellar observations in sophisticated research fields: the standard solar model (SSM), helio and asteroseismology, non-LTE 3D hydrodynamic photospheric modeling, nuclear reaction rates, solar neutrino observations, computational atomic physics, and plasma experiments. In this context, an unexpected downward revision of the solar photospheric metal abundances in 2005 spoiled a very precise agreement between the helioseismic indicators (the radius of the convection zone boundary, the sound-speed profile, and helium surface abundance) and SSM benchmarks, which could be somehow reestablished with a substantial opacity increase. Recent laboratory measurements of the iron opacity in physical conditions similar to the boundary of the solar convection zone have indeed predicted significant increases (30–400%), although new systematic improvements and comparisons of the computed tables have not yet been able to reproduce them. We give an overview of this controversy, and within the OP approach, discuss some of the theoretical shortcomings that could be impairing a more complete and accurate opacity accounting.

**Keywords:** opacity computations; opacity measurements; standard solar model; solar abundance problem; helioseismology; solar neutrinos

---

## 1. Introduction

The *Workshop on Astrophysical Opacities* [1] was held at the IBM Venezuela Scientific Center in Caracas during the week of 15 July 1991. Although this meeting included several leading researchers concerned with the computation of atomic and molecular opacities, it was a timely encounter between the two teams, namely the Opacity Project (OP) [1] and OPAL [2], that had addressed a 10-year-old plea [2] to revise the Los Alamos Astrophysical Opacity Library (LAAOL) [3]. The mood at the time was jovial because, in spite of the contrasting quantum mechanical frameworks and equations of state implemented by the two groups and the ensuing big-data computations, the general level of agreement of the new Rosseland mean opacities (RMO) was outstanding.

---

[1]   http://cdsweb.u-strasbg.fr/topbase/TheOP.html
[2]   http://opalopacity.llnl.gov/opal.html

Further refinements of the opacity data sets carried out the following decade, namely the inclusion of inner-shell contributions by the OP [4], led in fact to improved accord. However, a downward revision of the solar metal abundances in 2005 resulting from non-LTE 3D hydrodynamic simulations of the photosphere [5] compromised very precise benchmarks in the standard solar model (SSM) with the helioseismic and neutrino-flux predictions [6,7]. Since then things have never been the same.

Independent photospheric hydrodynamic simulations have essentially confirmed the lower metal abundances, specially of the volatiles (C, N, and O) [8]. The opacity increases required to compensate for the abundance corrections have neither been completely matched in extensive revisions of the numerical methods, theoretical approximations, and transition inventories [9–31], nor in a new generation of opacity tables (OPLIB) [3] by Los Alamos [32], nor even in recent innovative opacity experiments [33]. Nonetheless, unexplainable disparities in the modeling of asteroseismic observations of hybrid B-type pulsators with OP and OPAL opacities seem to still question their definitive reliability [34].

In the present report we give an overview of this ongoing multidisciplinary discussion, and within the context of the OP approach, analyze some of the theoretical shortcomings that could be impairing a more complete and accurate opacity accounting. In particular we discuss spectator-electron processes responsible for the broad and asymmetric resonances arising from core photoexcitation, K-shell resonance widths, and ionization edges since it has been recently shown that the solar opacity profile is sensitive to the Stark broadening of K lines [27].

## 2. OP and OPAL Projects

In a plasma where local thermodynamic equilibrium (LTE) can be assumed, the specific radiative intensity $I_\nu$ is not very different from the Planck function $B_\nu(T)$ and the flux can be estimated by a diffusion approximation

$$\mathbf{F} = -\frac{4\pi}{3\kappa_R}\frac{\mathrm{d}B}{\mathrm{d}T}\nabla T \tag{1}$$

with

$$\frac{\mathrm{d}B}{\mathrm{d}T} = \int_0^\infty \frac{\partial B_\nu}{\partial T}\mathrm{d}\nu \tag{2}$$

and

$$\frac{1}{\kappa_R} = \left[\int_0^\infty \frac{1}{\kappa_\nu}\frac{\partial B_\nu}{\partial T}\mathrm{d}\nu\right]\left[\frac{\mathrm{d}B}{\mathrm{d}T}\right]^{-1} . \tag{3}$$

Radiative transfer is then essentially controlled by the *Rosseland mean opacity*

$$\frac{1}{\kappa_R} = \int_0^\infty \frac{1}{\kappa_u} \times g(u)\mathrm{d}u \tag{4}$$

where the reduced photon energy is $u = h\nu/kT$ and the weighting function $g(u)$ is given by

$$g(u) = \frac{15}{4\pi^4}u^4\exp(-u)[1 - \exp(-u)]^{-2} . \tag{5}$$

The monochromatic opacity

$$\kappa_\nu = \kappa_\nu^{\mathrm{bb}} + \kappa_\nu^{\mathrm{bf}} + \kappa_\nu^{\mathrm{ff}} + \kappa_\nu^{\mathrm{sc}} \tag{6}$$

includes contributions from all the bound–bound (bb), bound–free (bf), free–free(ff), and photon scattering (sc) processes in the plasma. This implies massive computations of atomic data, namely energy levels (ground and excited), $f$-values, and photoionization cross sections for all the constituent ionic species of the plasma, and for a wide range of temperatures and densities, an adequate equation of state to determine

---

[3]   http://aphysics2.lanl.gov/opacity/lanl/

the ionization fractions and level populations in addition to the broadening mechanisms responsible for the line profiles.

## 2.1. Numerical Methods

In the computations of the vast atomic data required for opacity estimates—namely level energies, $f$-values, and photoionization cross sections—OP and OPAL used markedly different quantum-mechanical methods. The former adopted a multichannel framework based on the close-coupling expansion of scattering theory [35], where the $\Psi(SL\pi)$ wave function of an $(N+1)$-electron system is expanded in terms of the $N$-electron $\chi_i$ core eigenfunctions

$$\Psi(SL\pi) = \sum_i \chi_i \theta_i + \sum_j c_j \Phi_j(SL\pi) . \tag{7}$$

The second term in Equation (7) is a configuration-interaction (CI) expansion built up from core orbitals to account for orthogonality conditions and to improve short-term correlations. Core orbitals were generated with standard CI atomic structure codes such as SUPERSTRUCTURE [36] and CIV3 [37], and the total system wave functions and radiative data were computed with the $R$-matrix method [38,39] including extensive developments in the asymptotic region specially tailored for the project [40]. The latter allowed the treatment of both the discrete and continuum spectra for an ionic species in a unified manner (see Figure 1), its effectiveness depending on the accuracy and completeness of the core eigenfunction expansion in Equation (7), particularly at the several level crossings that take place along an isoelectronic sequence as the atomic number $Z$ increases.

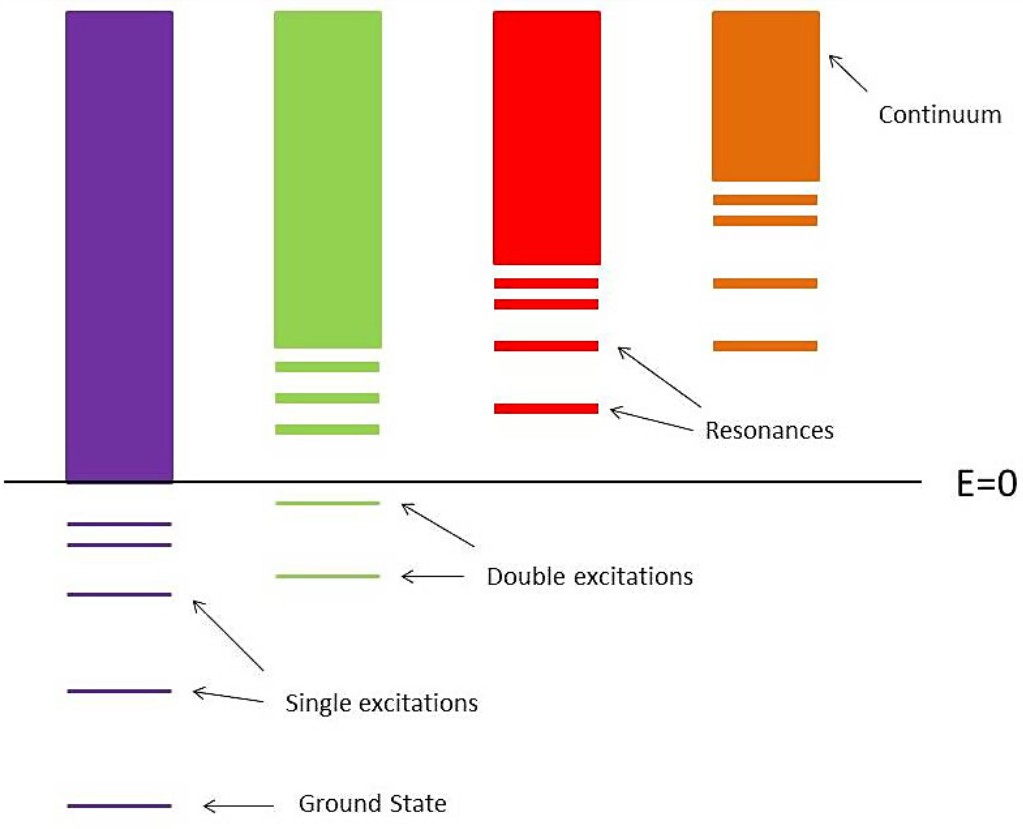

**Figure 1.** Multichannel ionic structure showing series of singly and doubly excited bound states and resonances and several continua.

The OPAL code, on the other hand, relied on parametric potentials that generated radiative data of accuracy comparable to single-configuration, self-consistent-field, relativistic schemes [41,42]. The parent (i.e., core) configuration defines a Yukawa-type potential

$$V = -\frac{2}{r}\left((Z-v) + \sum_{n=1}^{n_{\max}} N_n \exp(-\alpha_n r)\right) \qquad (8)$$

with $v = \sum_{n=1}^{n_{\max}} N_n$ for all the subshells and scattering states available to the active electron. In Equation (8), $N_n$ is the number of electrons in the shell with principal quantum number $n$, $n_{\max}$ being the maximum value in the parent configuration, and $\alpha_n$ are $n$-shell screening parameters determined from matches of solutions of the Dirac equation with spectroscopic one-electron, configuration-averaged ionization energies. Thus, in this approach the atomic data were computed on the fly as required rather than relying on stored files (as in OP) that might prove inadequate in certain plasma conditions.

*2.2. Equation of State*

The formalisms of the equation of state implemented by OP and OPAL were also significantly different. In the OP the chemical picture was emphasized [43,44], where clusters (i.e., atoms or ions) of fundamental particles can be identified and the partition function of the canonical ensemble is factorizable. Thermodynamic equilibrium occurs when the Helmholtz free energy reaches a minimum for variations of the ion densities, and the internal partition function for an ionic species $s$ can then be written as

$$Z_s = \sum_i w_{is}\, g_{is} \exp(-E_{is}/kT) \qquad (9)$$

where $g_{is}$ and $w_{is}$ are respectively the statistical weight and occupation probability of the $i$th level. The main problem was then the estimate of the latter, which was performed assuming a nearest-neighbor approximation for the ion microfield or by means of distribution functions that included plasma correlation effects.

The OPAL equation of state was based on many-body activity expansions of the grand canonical partition function that avoided Helmholtz free-energy compartmentalization [45]. Since pure Coulomb interactions between the electrons and nuclei are assumed, the divergence of the partition function does not take place; thus, there is no need to consider explicitly the plasma screening effects on the bound states.

*2.3. Revised Opacities*

When compared to the LAAOL (see Figure 2), preliminary OPAL iron monochromatic opacities at density $\rho = 6.82 \times 10^{-5}\,\mathrm{g\,cm^{-3}}$ and temperature $T = 20$ eV indicated large enhancements at photon energies around 60 eV [46]. They are due to an $n = 3 \rightarrow 3$ unresolved transition array that significantly enhances the Rosseland mean since its weighting function peaks at around 80 eV. Within the OP $R$-matrix approach, such transitions were computationally intractable at the time, and were then treated for the systems Fe VIII to Fe XIII in a simpler CI approach (i.e., second term of Equation (7)) with the SUPERSTRUCTURE atomic structure code [47]. The dominance of this $\Delta n$ unresolved transition array in Fe was subsequently verified by laboratory photoabsorption measurements [48].

It was later pointed out that, in spite of general satisfactory agreement, the larger differences between the OPAL and OP opacities were at high temperatures and densities due to missing inner-shell contributions in OP [49]. Hence, the latter were systematically included in a similar CI approach with the AUTOSTRUCTURE code in what became a major updating of the OP data sets [4,50,51] illustrated in Figure 3 (left) for the solar S92 mix. After this effort and as shown in Figure 3 (right), the OPAL and OP opacities were thought to be in good working order, and it was actually mentioned that they could not be differentiated in stellar pulsation calculations [52].

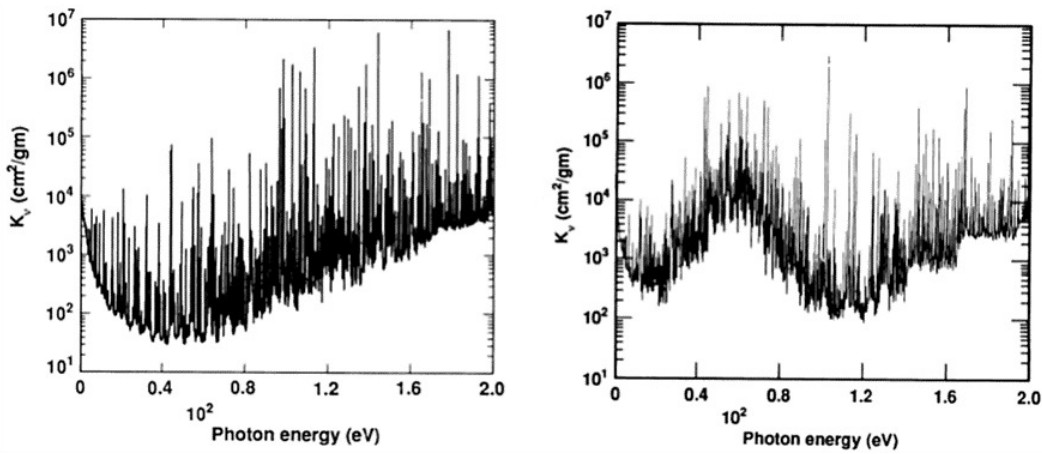

**Figure 2.** Monochromatic Fe opacities at $\rho = 6.82 \times 10^{-5}\,\mathrm{g\,cm^{-3}}$ and $T = 20$ eV. **Left** panel: LAAOL. **Right** panel: OPAL. Reproduced from Figures 1 and 2 of [46] with permission of the ©AAS and the Lawrence Livermore National Laboratory.

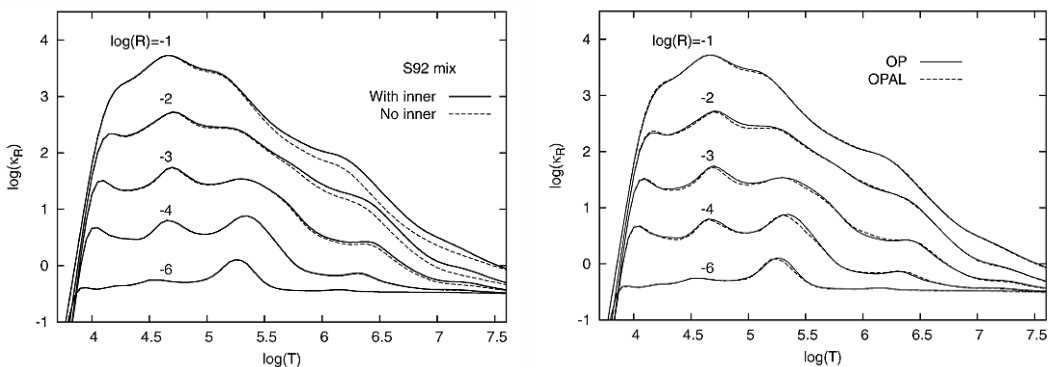

**Figure 3.** **Left**: Inner-shell contributions to the Rosseland-mean opacities for the S92 mix. **Right**: The OP and OPAL Rosseland-mean opacities for the S92 mix. Note: $R = \rho/T_6^3$ where $\rho$ is the mass density in $\mathrm{g\,cm^{-3}}$ and $T_6 = 10^{-6} \times T$ with $T$ in K. Reproduced from Figures 1 and 2 of Ref. [4].

### 2.4. Big-Data Science

The OP was a pioneering example of what is now called collaborative big-data science [53]. It involved the computation of large data sets by several internationally distributed research groups, which were then compiled and stringently curated before becoming part of what eventually became TOPbase [54,55], one of the first online atomic databases. In order to ensure machine independence, its database management system was completely developed from scratch in standard Fortran-77, and its user interface rapidly evolved from command-based to web-browser-based. Furthermore, TOPbase was housed right from the outset in a *data center*, namely the Centre de Données astronomiques de Strasbourg (CDS) [4], also a seminal initiative at the time.

Regarding the dissemination of monochromatic and mean opacities for arbitrary chemical mixes, the OP also implemented the innovative concept of a *web service*, namely the OPserver [56], accessible online from the Ohio Supercomputer Center [5]. Since the main overhead in the calculation of mean opacities or radiative accelerations is the time taken to read large data volumes from secondary storage (disk), the OPserver always keeps the bulk of the monochromatic opacities in

---

4　http://cdsweb.u-strasbg.fr/
5　https://www.osc.edu/

main memory (RAM) waiting for a service call from a web portal or the modeling code of a remote user; furthermore, the OPserver can also be downloaded (data and software) to be installed locally in a workstation. Its current implementation and performance are tuned up to streamline stellar structure or evolution modeling requiring mean opacities for varying chemical mixtures at every radial point or time interval.

## 3. Solar Abundance Problem

### 3.1. Standard Solar Model

In 2005 there was much expectation in the solar modeling community for the OP inner-shell opacity update (see Section 2.3) as the very precise and much coveted benchmarks of the SSM with the helioseismic indicators—namely the radius of the convection zone boundary (CZB), He surface abundance, and the sound-speed profile—had been seriously disrupted by a downward revision of the photospheric metal abundances [5]. This was was the result of advanced 3D hydrodynamic simulations that took into account granulation and non-LTE effects [6,7]. In this regard, the final improved agreement between the OP and OPAL mean opacities after the aforementioned update turned out to be a disappointment in the SSM community.

This situation can be appreciated in Table 1, where the SSM benchmarks with the helioseismic measurements of the CZB and He surface mass fraction are hardly modified by the opacity choice (OPAL or OP), while the new abundances [5] lead to noticeable modifications. It is further characterized by the SSM discrepancies with the helioseismic sound-speed profile near the CZB (see Figure 4), which has been shown to disappear with an opacity increase of $\sim 30\%$ in this region down to a few percent in the solar core [57].

**Table 1.** Standard solar model (SSM) benchmarks (BS05 model from [7]) with the helioseismic predictions [58], namely the depth of the CZB ($R_{cz}/R_{sun}$) and He surface mass fraction ($Y_{sur}$), using OP and OPAL opacities and the standard (GS98) [59] and revised (AGS05) [5] metal abundances.

| Model | $R_{cz}/R_{sun}$ | $Y_{sur}$ |
|---|---|---|
| BS05(GS98,OPAL) | 0.715 | 0.244 |
| BS05(GS98,OP) | 0.714 | 0.243 |
| BS05(AGS05,OPAL) | 0.729 | 0.230 |
| BS05(AGS05,OP) | 0.728 | 0.229 |
| Helioseismology | 0.713(1) | 0.249(3) |

### 3.2. Seismic Solar Model

The sound-speed profile in the solar core has also been useful, in what is referred to as the seismic solar model (SeSM) [60], to verify the reliability of the highly temperature-dependent (and thus opacity-dependent) nuclear reaction rates in reproducing the observed solar neutrino fluxes. In other words, both the helioseismic indicators and neutrino fluxes impose fairly strict constraints on the opacities and metal abundances (both of volatile and refractory elements) at different solar radii. For instance, in spite of improved accord with the helioseismic CZB depth and sound speed, the higher metallicity recently derived from in situ measurements of the solar wind [61] (see Table 2) has been seriously questioned [62–64] because of a neutrino overproduction caused by the refractory excess (i.e., from Mg, Si, S, and Fe). In Table 2, a more recent revision of the photospheric metallicity [65] is also tabulated that is $\sim$10% higher but still $\sim$25% lower than the previously assumed standard [59], and which has been independently confirmed by a similar 3D hydrodynamic approach [8]. It has been shown [66] that, with the more recent abundances of [65], the required opacity increase in the SSM to restore the helioseismic benchmarks is only now around 15% at the CZB and 5% in the core, i.e., half of what was previously estimated [57].

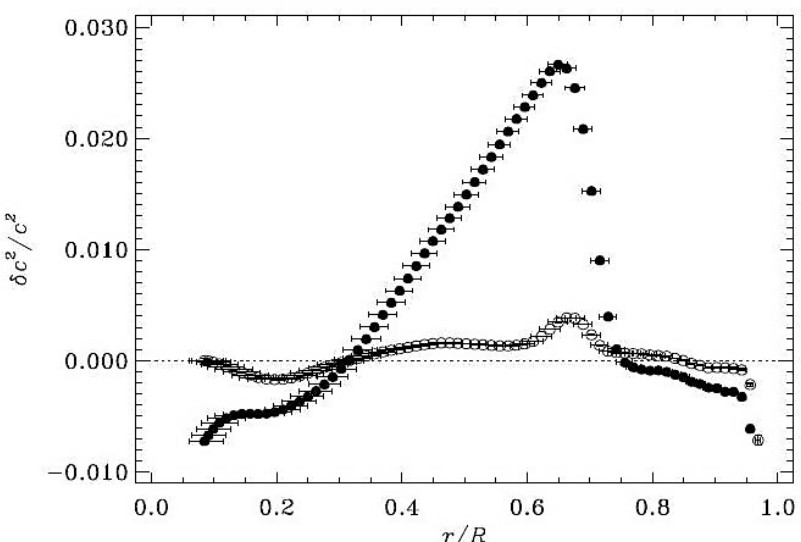

**Figure 4.** Relative sound-speed differences between helioseismic measurements and SSM using the standard solar composition [59] (open circles) and the revised metal abundances [5] (filled circles). Reproduced from Figure 1 of [57] with permission of ©ESO.

**Table 2.** Solar abundances ($\log \epsilon_i \equiv \log N_i / N_{\mathrm{H}} + 12$). GS98: [59]. AGS05: [5]. AGSS09: [65]. CLSFB11: [8]. SZ16: [61].

| $i$ | GS98 | AGS05 | AGSS09 | CLSFB11 | SZ16 |
|---|---|---|---|---|---|
| C | 8.52 | 8.39(5) | 8.43(5) | 8.50(6) | 8.65(8) |
| N | 7.92 | 7.78(6) | 7.83(5) | 7.86(12) | 7.97(8) |
| O | 8.83 | 8.66(5) | 8.69(5) | 8.76(7) | 8.82(11) |
| Ne | 8.08 | 7.84(6) | 7.93(10) | | 7.79(8) |
| Mg | 7.58 | 7.53(9) | 7.60(4) | | 7.85(8) |
| Si | 7.56 | 7.51(4) | 7.51(3) | | 7.82(8) |
| S | 7.20 | 7.14(5) | 7.12(3) | 7.16(5) | 7.56(8) |
| Fe | 7.50 | 7.45(5) | 7.50(4) | 7.52(6) | 7.73(8) |
| $Z/X$ | 0.0229 | 0.0165 | 0.0181 | 0.0209 | 0.0265 |

### 3.3. Multidisciplinary Approach

Consensus has been reached in a multidisciplinary community encompassing sophisticated research fields (see Figure 5), for which the reliability of computed and measured of opacities is a key concern, that there is at present a solar abundance problem. This critical situation is encouraging further developments and discussions that, in a similar manner to the solar neutrino problem, are expected to converge soon to a comprehensive solution. For instance, since the SSM is basically a static representation, there has been renewed interest in considering magnetic and dynamical effects such as rotation, elemental diffusion, and mixing and convection overshoot [60,67]. Nuclear fusion cross sections have also been critically evaluated pinpointing cases for which the current precision could be improved [68].

Helioseismology has become a powerful diagnostic technique; for instance, it remarkably enables the performance evaluation of the different equations of state [69], and with the advent of the *Kepler* [6] and *CoRoT* [7] space probes, it has been successfully applied to other stellar structures and evolutionary

---

[6]   http://kepler.nasa.gov/
[7]   http://www.esa.int/Our_Activities/Space_Science/COROT

models in what is rapidly becoming a new well-established research endeavor: asteroseismology [70]. In this regard, it has been curiously shown that, in the complex asteroseismology of some hybrid B-type pulsators (e.g., $\gamma$ Pegasi), both OPAL and OP opacity tables perform satisfactorily [71], but in others ($\nu$ Eridani) inconsistencies appear in the analysis that still seem to indicate missing opacity [34], a point that is treated to some extent in Sections 4 and 5.

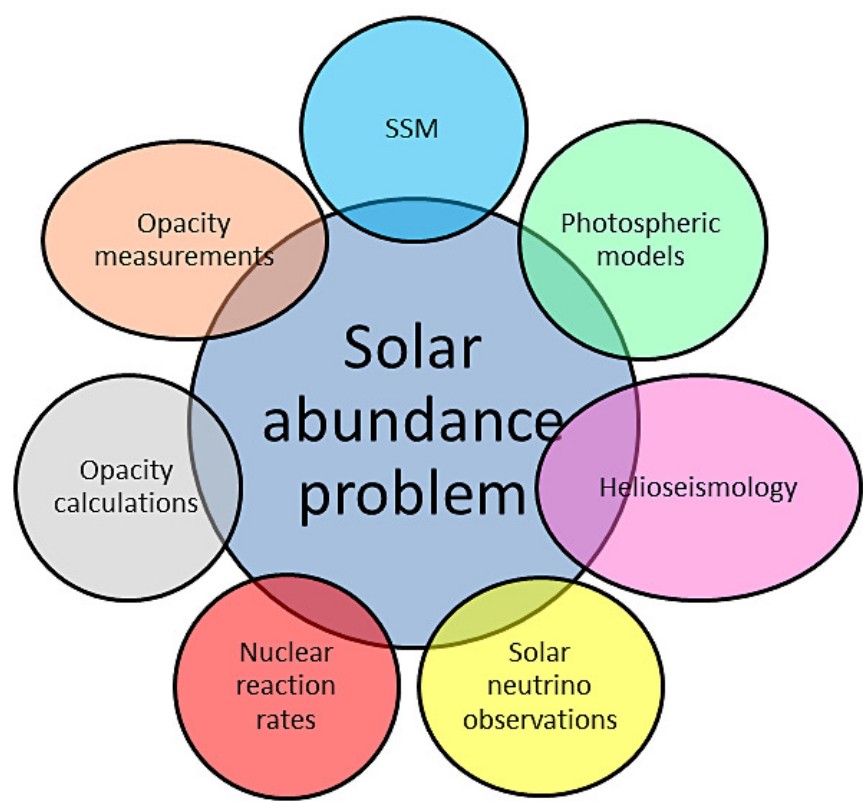

**Figure 5.** Research fields of the multidisciplinary community involved in the solar abundance problem.

## 4. Atomic Opacities, Recent Developments

In a similar way to other research areas of Figure 5, the solar abundance problem has stimulated detailed revisions and new developments of the astrophysical opacity tables and the difficult laboratory reproduction of the plasma conditions at the base of the solar convection zone to obtain comparable measurements.

The OPAC international consortium [14,17] has been carrying out extensive comparisons using a battery of opacity codes (SCO, CASSANDRA, STA, OPAS, LEDCOP, OP, and SCO-RCG), looking into the importance of configuration interaction and accounting mainly in elements of the iron-group bump ($Z$ bump), i.e., Cr, Fe, and Ni, due to their relevance in the pulsation of intermediate-mass stars. In this respect, the magnitude of the Fe and Ni contributions to the $Z$-bump and the temperature at which they occur have been shown to be critical in p- and g-mode pulsations in cool subdwarf B stars [72,73]. Problems with both the OPAL and OP tables have been reported; for instance, there are considerable differences in the OPAS and OP single-element monochromatic opacities even though the OPAS RMO $\kappa_R$ for the solar mixture by [59] agree to within 5% with both OPAL and OP for the entire radiative zone of the Sun ($0.0 \leq r/R_\odot \leq 0.713$) [13]. The larger $\kappa_R^{OPAS}/\kappa_R^{OP}$ ratios are found in Mg ($-44\%$), Fe ($+40\%$), and Ni ($+47\%$) due to variations in the ionic fractions (particularly for the lower charge states), missing configurations in OP, and the pressure broadening of the K$\alpha$ line in the He-like systems.

A new generation of Los Alamos opacity tables (OPLIB) including elements with atomic number $Z \leq 30$ have been computed with the ATOMIC code and made publicly available [32]. They have

been used to study the $Z$ bump, finding reasonable agreement with transmission measurements in the XUV but factors of difference with the OP RMO tables for Fe and Ni [31]. Additionally, relativistic opacities for Fe and Ni computed with the Los Alamos codes are found to be in good agreement with semi-relativistic versions bolstering confidence in their numerical methods [21].

Solar models have been recently computed with both the OPLIB and OPAL opacity tables and the metal abundances of [65], finding only small differences in the helioseismic indicators although the OPLIB data display steeper opacity derivatives at the temperatures and densities associated with the solar interior [74]. These opacity derivatives directly impact stellar pulsation properties such as the driving frequency, and since the latter data set has also been shown to yield a higher RMO than OPAL and OP in the $Z$-bump region, it gives rise to wider B-type-pulsator instability domains [75]. However, basic pulsating star problems remain unsolved such as the pulsations of the B-type stars in the Large and Small Magellanic Clouds. Moreover, in a recent analysis of the oscillation spectrum of $\nu$ Eridani with different opacity tables, OPLIB is to be preferred over OPAL and OP, but the observed frequency ranges can only be modeled with substantially modified mean-opacity profiles (an increase of a factor of 3 or larger at $\log(T) = 5.47$ to ensure g-mode instability and a reduction of 65% at $\log(T) = 5.06$ to match the empirical $f$ non-adiabatic parameters) that are nevertheless impaired by puzzling side effects: enhanced convection efficiency in the $Z$ bump affecting mode instability and avoided-crossing effects in radial modes [76].

The OPAL opacity code has been updated (now referred to as TOPAZ) to improve atomic data accuracy and used to recompute the monochromatic opacities of the iron-group elements [22]; small decreases (less than 6%) with respect to the OPAL96 tables have been found. The OP collaboration (IPOPv2), on the other hand, has been performing test calculations on the Ni opacity by treating Ni XIV as a study case [16], and a new online service for generating opacity tables is available [77].

To bypass the computational shortcomings of the detailed-line-accounting methods in managing the huge number of spectral lines of the more complex ions, novel methods based on variations and extensions of the *unresolved-transition-array* (UTA) concept [78] have been implemented. In these methods, the transition array between two configurations is usually reduced to a single Gaussian but can be replaced by a sum of partially resolved transition arrays represented by Gaussians [79,80] that are then sampled statistically to simulate detailed line accounting [81]. For heavy ions where UTAs still fall short, a higher-level extension referred to as the *super-transition-array* (STA) method involves the grouping of many configurations into a single superconfiguration, whereby the UTA summation to a single STA can be performed analytically [82–84]. Recent STA calculations of the RMO in the solar convection zone boundary agree very well with both OP and OPAL, and predict a meager heavy-element ($Z > 28$) contribution [26].

The most salient effort has perhaps been the recent measurements of Fe monochromatic opacities in laboratory plasma conditions similar to the solar CZB ($T_e = 1.9-2.3$ MK and $N_e = 0.7-4.0 \times 10^{22}$ cm$^{-3}$) that show increments (30–400%) that have not yet been possible to match theoretically [33]. However, the mean-opacity enhancements are still not large enough (only 50%) to restore the SSM helioseismic benchmarks, but sizeable experimental increases in other elements of the solar mixture such as Cr and Ni would certainly reduce the current discrepancy. Furthermore, a point of concern in this experiment is the model dependency of the electron temperature and density determined by K-shell spectroscopy of tracer Mg; however, as discussed in [85], it is not expected to be large enough to explain the standing disaccord with theory.

The line broadening approximations implemented in most opacity calculations have been recently questioned [27], reporting large discrepancies between the OP K-line widths and those in other opacity codes. It is also shown therein that the solar opacity profile is sensitive to the pressure broadening of K lines, which can be empirically matched with the helioseismic indicators by a K-line width increase of around a factor of 100. This moderate line-broadening dependency (a few percent) of the solar opacity near the bottom of the convection zone concurs with previous findings [51,86].

### 4.1. Fe Opacity

The iron opacity is without a doubt the most revered in atomic astrophysics due, on the one hand, to its relevance in the structure of the solar interior and in the driving of stellar pulsations, and on the other, to the difficulties in obtaining accurate and complete radiative data sets for the Fe ions. In this respect, the versatile HULLAC relativistic code [87] has been used to study CI effects mainly in $3 \rightarrow 3$ and $3 \rightarrow 4$ transitions that dominate the maximum of the RMO [19]. At $T = 27.3$ eV and $\rho = 3.4$ mg cm$^{-3}$, it is found that CI causes noticeable changes in the spectrum shape due to line shifts at the lower photon energies; good agreement is found with OP in contrast to the spectra by other codes such as SCO-RCG, LEDCOP, and OPAS. The importance of CI is also brought out in comparisons of the OP and OPAL monochromatic opacities with old transmission measurements, namely spectral energy displacements [19]. However, a further comparison [31] with recent results at $T = 15.3$ eV and $\rho = 5.48$ mg cm$^{-3}$ obtained with the ATOMIC modeling code, results that include contributions from transitions with $n \leq 5$, shows significantly higher monochromatic opacities than OP for photon energies greater than 100 eV and almost a factor of 2 increase in the RMO; it is pointed out therein that this is due to limited M-shell configuration expansions in OP for ions such as Fe VIII.

A more controversial situation involves the recently measured Fe monochromatic opacities that proved to be higher that expected in conditions similar to the solar CZB [33]. Such a result came indeed as a surprise since previous comparisons of transmission measurements at $T = 156 \pm 6$ eV and $N_e = (6.9 \pm 1.7) \times 10^{21}$ cm$^{-3}$ with opacity models—namely ATOMIC, MUTA, OPAL, PRISMSPECT, and TOPAZ—showed satisfying line-by-line agreement [32,88,89]. Relative to [33], the OP values are not only lower, but the wavelengths of the strong spectral features are not reproduced accurately. Other codes, namely ATOMIC, OPAS, SCO-RCG, and TOPAZ, perform somewhat better regarding the positions of the spectral features but the absolute backgrounds are also generally too low; moreover, the computed RMO are in general smaller than experiment by factors greater than 1.5. It has been suggested that calculations have missing bound–bound transitions or underestimate certain photoionization cross sections [33]. The measured windows are higher than those predicted by models, the peak values show a large (50%) scatter, and the measured widths of prominent lines are significantly broader. On the other hand, from the point of view of oscillator-strength distributions and sum rules, the measured data appear to display unexplainable anomalies [89].

A recent *R*-matrix calculation [28] involving the topical Fe XVII ionic system has found large (orders of magnitude) enhancements in the background photoionization cross sections and huge asymmetric resonances produced by core photoexcitation that lead to higher (35%) RMO; however, such claims have been seriously questioned [8] [29,30,90].

### 4.2. Ni and Cr Opacities

Due to their contributions to the Z bump, the Ni and Cr opacities have also received considerable attention. For the former, it has been shown [19] that CI effects on the RMO are not as conspicuous as those of Fe: they manifest themselves mainly at lower photon energies (50–60 eV), but the spectral features are distinctively different from those of the OP; the latter are believed to be faulty as they were determined by extrapolation procedures. A similar diagnostic has been put forward for Cr, and both are supported by recent transmission measurements in the XUV [31]; moreover, comparisons with data from the ATOMIC code apparently lead to discrepancies with the OP Ni RMO of a factor of 6.

## 5. Line, Resonance, and Edge Profiles

As mentioned in Section 4, a study of line profiles in opacity calculations has shown that OP K-line widths are largely discrepant with those in other plasma models in conditions akin to the

---

[8]    A. K. Pradhan, private communication (2017).

solar CZB [27]. Such a finding in fact reflects essential differences in the treatment of ionic models in opacity calculations; i.e., between the OP multichannel representations (see Figure 1) and the simpler uncoupled, or in some cases statistical, versions adopted by other projects that become particularly conspicuous in spectator-electron processes. The latter are responsible for K-line widths, the broad profiles of resonances resulting from core photoexcitation (PECs hereafter) and ionization-edge structures. The K-line pressure broadening issue is also examined.

### 5.1. Spectator-Electron Processes

It has been shown that the electron impact broadening of such features is intricate, requiring the contributions from the interference terms that cause overlapping lines to coalesce such that the spectator-electron relaxation does not affect the line shape, i.e., narrower lines with restricted wings [91]. Electron impact broadening of resonances in the close-coupling formalism is also currently under review [9]. Furthermore, as discussed by [24], spectator-electron transitions can give rise to a large number of X-ray satellite lines that can significantly broaden the resonance-line red wing; since they are difficult to treat in the usual detailed-line-accounting approach, the approximate statistical methods based on UTAs [79–81] previously mentioned (see Section 4) have been proposed.

In Sections 5.1.1–5.1.3 we briefly discuss how such processes are currently handled within the *R*-matrix framework, in particular with regard to the problem of complete configuration accounting.

### 5.1.1. PEC Resonances

PECs were first discussed in the OP by [92], and to illustrate their line shapes and widths, we consider the photoabsorption of the $3sns\,^1S$ states of Mg-like Al II described in [93]:

$$3sns\,^1S + \gamma \rightarrow 3pn's\,^1P^o \rightarrow 3s\,^2S + e^- . \tag{10}$$

The PEC arises when $n = n'$ wherein the $ns$ active electron does not participate in the transition. It may be seen in Figure 6a that the photoabsorption cross section of the Al II $3s^2$ ground state is dominated by a series of $3pn's$ asymmetric window resonances without a PEC since, for $n = n' = 3$, $3s3p$ is a true bound state below the ionization threshold. This is not the case for $3s4s\,^1S$, the first excited state (see Figure 6b), where now the $3p4s$ resonance lying just above the threshold becomes a large (two orders of magnitude over the background) PEC. A similar situation occurs in the photoabsorption cross sections of the following excited states of the series, namely $3s5s$ and $3s6s$, that are respectively dominated by the $3p5s$ and $3p6s$ PECs (see Figures 6c,d). It may be appreciated that the $3pns$ PEC widths are approximately constant and independent of the $n$ principal quantum number since their oscillator-strength distributions are mostly determined by the $f(3s, 3p) = 0.849$ oscillator strength of the Na-like Al III core [94].

A further relevant point is that, to obtain the $3pns$ PECs in the photoabsorption cross sections of the $3sns$ bound-state series, it suffices to include in the close-coupling expansion of Equation (7) the Na-like $3s$ and $3p$ core states; however, if the $4s$ and $4p$ states are additionally included, $4pns$ PECs will appear in the cross sections of the $3sns$ excited states for $n \geq 4$, whose resonance properties would be dominated by the Al III core $f(3s, 4p) = 0.0142$ oscillator strength [94]. Due to the comparatively small value of the latter, such PECs will be less conspicuous, but in systems with more complicated electron structures this will not in general be the case. In conclusion, the PEC $n$ inventory directly depends on the $n_{\max}$ of core-state expansion in Equation (7) whose convergence would unavoidably lead to large calculations.

---

[9]    A. K. Pradhan, private communication (2016).

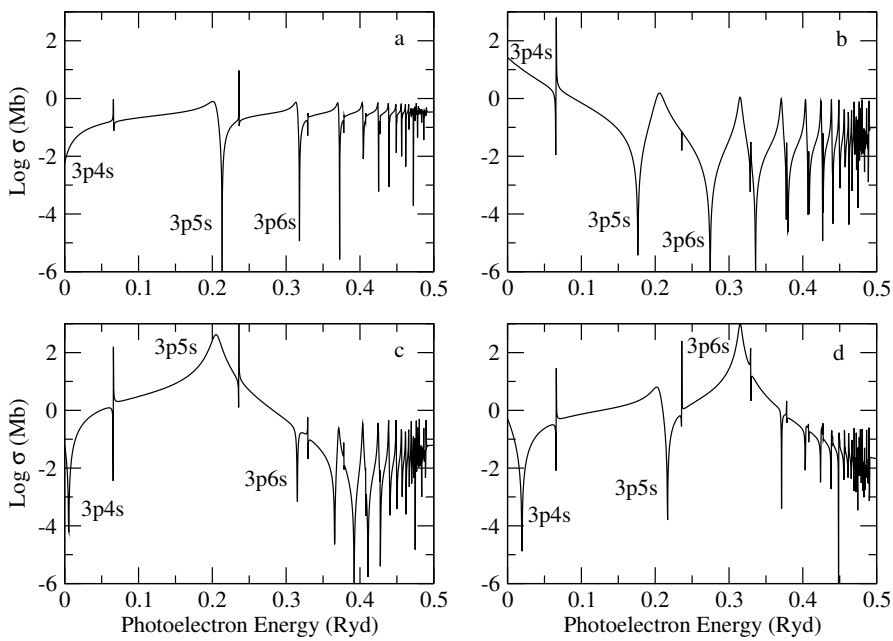

**Figure 6.** Photoabsorption cross sections of the $3sns$ states of Al II showing the $3pns$ PEC resonances for $n \geq 4$. (**a**) $3s^2$; (**b**) $3s4s$; (**c**) $3s5s$; (**d**) $3s6s$.

### 5.1.2. K Lines

K-resonance damping is another example of the dominance of spectator-electron processes, which can be illustrated with the photoexcitation of the ground state of Fe XVII to a K-vacancy Rydberg state

$$h\nu + 1s^2 2s^2 2p^6 \longrightarrow 1s 2s^2 2p^6 np \,. \tag{11}$$

This state decays via the radiative and Auger manifold

$$1s 2s^2 2p^6 np \xrightarrow{\text{K}n} 1s^2 2s^2 2p^6 + h\nu_n \tag{12}$$

$$\xrightarrow{\text{K}\alpha} 1s^2 2s^2 2p^5 np + h\nu_\alpha \tag{13}$$

$$\xrightarrow{\text{KL}n} \begin{cases} 1s^2 2s^2 2p^5 + e^- \\ 1s^2 2s 2p^6 + e^- \end{cases} \tag{14}$$

$$\xrightarrow{\text{KLL}} \begin{cases} 1s^2 2s^2 2p^4 np + e^- \\ 1s^2 2s 2p^5 np + e^- \\ 1s^2 2p^6 np + e^- \end{cases} , \tag{15}$$

which has been shown to be dominated by the radiative K$\alpha$ (Equation (13)) and Auger KLL (Equation (15)) spectator-electron channels [95]. Such damping process causes the widths of the $1s 2s^2 2p^6 np$ K resonances to be broad, symmetric, and almost independent of $n$, leading to the smearing of the K edge as shown in Figure 7.

An inherent difficulty of the *R*-matrix method is that, to properly account for the damped widths of K vacancy states such as $1s 2s^2 2p^6 np$, it requires the inclusion of the $1s^2 2s^2 2p^4 np$, $1s^2 2s 2p^5 np$, and $1s^2 2p^6 np$ core configurations of the dominant KLL channels (see Equation (15)) in the close-coupling expansion, which would rapidly make the representation of the high-$n$ resonances as $n \to \infty$ computationally intractable. To overcome this limitation, Auger damping is currently managed within the *R*-matrix formalism by means of an optical potential [96], which however requires the determination of the core Auger widths beforehand with an atomic structure code (e.g., AUTOSTRUCTURE). Furthermore, since K lines have such distinctive symmetric profiles,

this scheme of predetermining made-to-order damped widths to be then included in opacity calculations could also be easily implemented in perturbative methods.

Following [95], the estimated opacity of a photoionized gas with solar elemental abundances and ionization parameter $\xi = 10$ is depicted in Figure 8 in the photon range 7–8 KeV adopting damped and undamped cross sections. A larger number of broad peaks, particularly the K$\alpha$ transition array around 7.2 KeV, and smeared edges are distinctive damping features. (It must be mentioned that resonances in Figures 7 and 8 are not fully resolved, so in some cases their peak values may appear to be underestimated suggesting departures from line-strength preservation.)

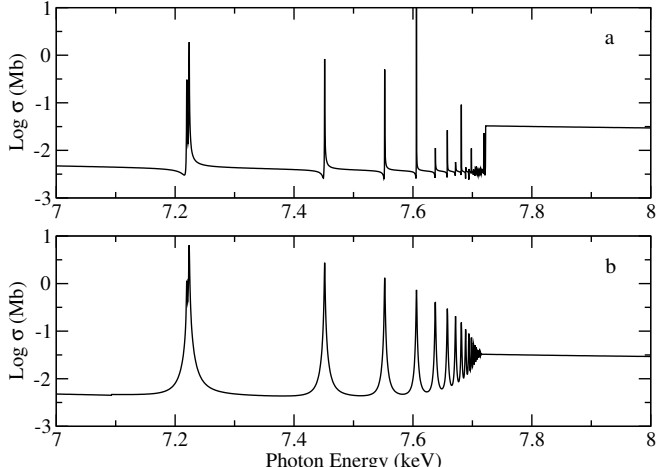

**Figure 7.** Total K photoabsorption cross section of the $1s^2 2s^2 2p^6\,^1S$ ground state of Fe XVII. (**a**) Undamped cross section. (**b**) Damped (radiation and Auger) cross section.

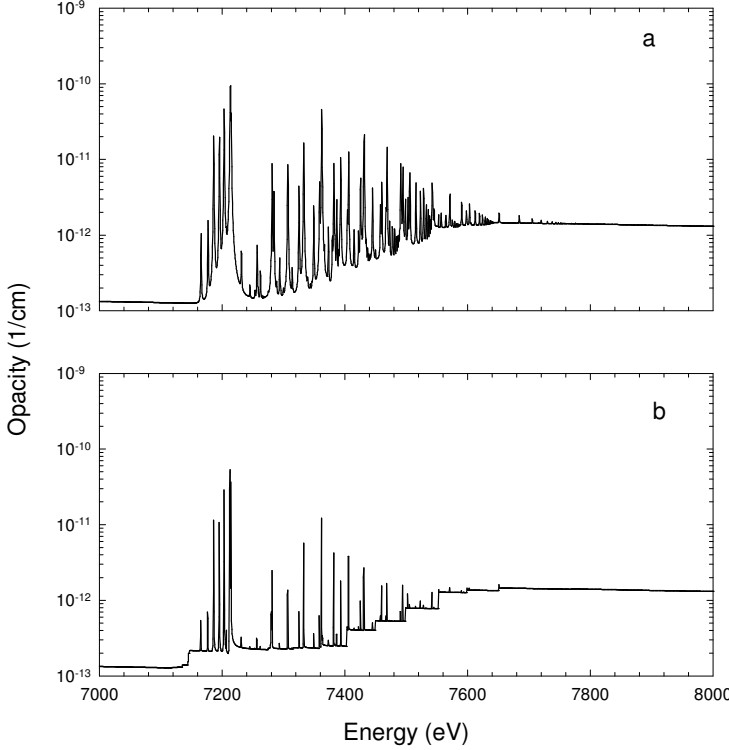

**Figure 8.** Monochromatic opacity of a photoionized gas with solar abundances and ionization parameter $\xi = 10$, (**a**) estimated with damped cross sections and (**b**) estimated with undamped cross sections. Reproduced from Figure 3 of [95] with permission of the ©AAS.

### 5.1.3. Ionization Edges

While K and L ionization edges are well understood in ground-state photoabsorption cross sections in the *R*-matrix formalism, those for excited states have not received comparable attention. In the same way, as PECs (see Section 5.1.1), the K edge arises from a spectator-electron transition whose adequate representation is limited by the $n_{max}$ of the core-state CI complex in the close-coupling expansion. This assertion may be illustrated using the simple O VI Li-like system as a study case.

Let us consider K photoabsorption of the $1s^2ns$ series of this ion; the K edge occurs as

$$1s^2ns + \gamma \longrightarrow 1sns + e^-(kp) \tag{16}$$

where the *ns* electron remains a spectator in the transition; therefore, in a similar fashion to PECs, the K-edge *n*-inventory would depend on the $n_{max}$ of the core-state representation in the close-coupling expansion (Equation (7)). In Figure 9, the photoabsorption cross sections of these levels are plotted for $n \leq 4$ using three target representations for the He-like core:

**Target A** —$1s^2$, $1s2\ell$ with $\ell \leq 1$
**Target B** —Target A plus $1s3\ell'$ with $\ell' \leq 2$
**Target C** —Target B plus $1s4\ell''$ with $\ell'' \leq 3$.

It may be seen that, when Target A ($n_{max} = 2$) is implemented, a K edge appears in the photoabsorption cross section of the $1s^22s$ level but not in those of $1s^23s$ and $1s^24s$; for Target B, where $n_{max} = 3$, K edges are there except in the cross section of the latter; and for Target C with $n_{max} = 4$, they are all present. As shown in Figure 9, adequate K-edge representations for excited states are essential to obtain accurate high-energy tails in their cross sections.

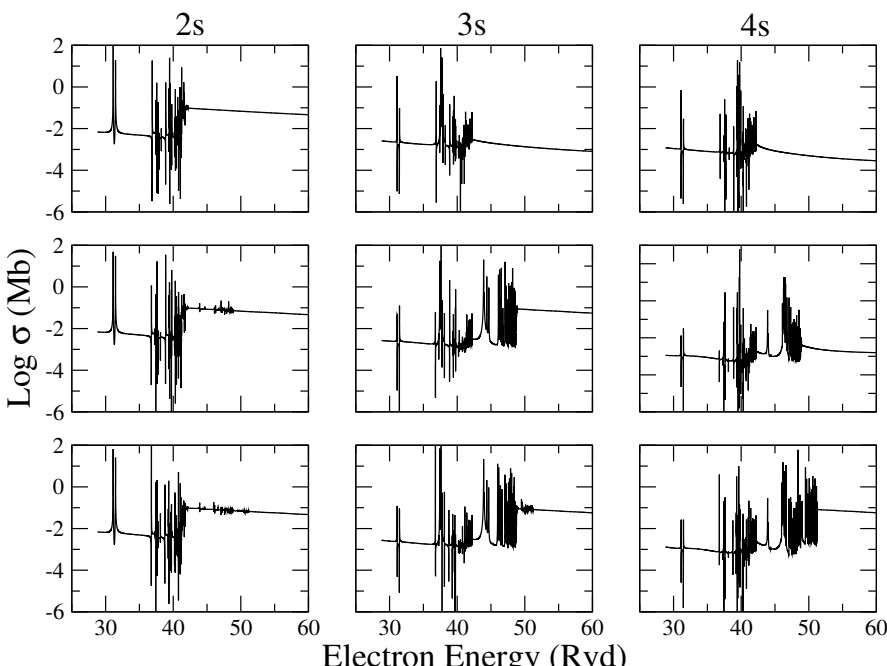

**Figure 9.** K photoabsorption cross sections of the $1s^2ns$ states of O VI. Left column: 2*s* state. Middle column: 3*s* state. Right column: 4*s* state. First row: computed with Target A. Second row: computed with Target B. Third row: computed with Target C.

Furthermore, it may be seen in Figure 9 that, in the photoabsorption of the $1s^2ns$ states of O VI, the strong K lines are also the result of spectator-electron transitions of the type

$$1s^2ns + \gamma \longrightarrow 1s\,ns\,np \,. \tag{17}$$

That is, Target A would be sufficient to generate the Kα lines at photoelectron energies of ∼32 Ryd in the cross sections of the three states, but to obtain the broad Kβ transitions at ∼44 Ryd in the $1s^23s$ cross section, at least Target B is required. Target C would then be necessary to obtain the broad Kγ lines at ∼47 Ryd in the $1s^24s$ cross section. This conclusion implies that, in the *R*-matrix method, satellite lines must be explicitly specified in the close-coupling expansion. These difficulties of the *R*-matrix method in obtaining adequate target representations in spectator-electron processes, particularly when involving excited states, are at the core of the controversy around the extended *R*-matrix calculation of Fe XVII L-shell photoabsorption [28–30,90] mentioned in Section 4.1.

### 5.2. Pressure Broadening

To unravel the discrepancies of OP K-line Stark widths with those computed by other opacity codes, we adopt the simpler oxygen opacity as a test bed. This type of comparison has been previously carried out in [13,27,97]. In Figure 10 pure oxygen monochromatic opacities (corrected for stimulated emission) at $T = 192.91$ eV and $N_e = 10^{23}$ cm$^{-3}$ are plotted as a function of the reduced photon energy $u = h\nu/kT$, where the general overall agreement between OP [4] with OPLIB [97] and STAR [27] (see upper panel) is very good except for the line wings.

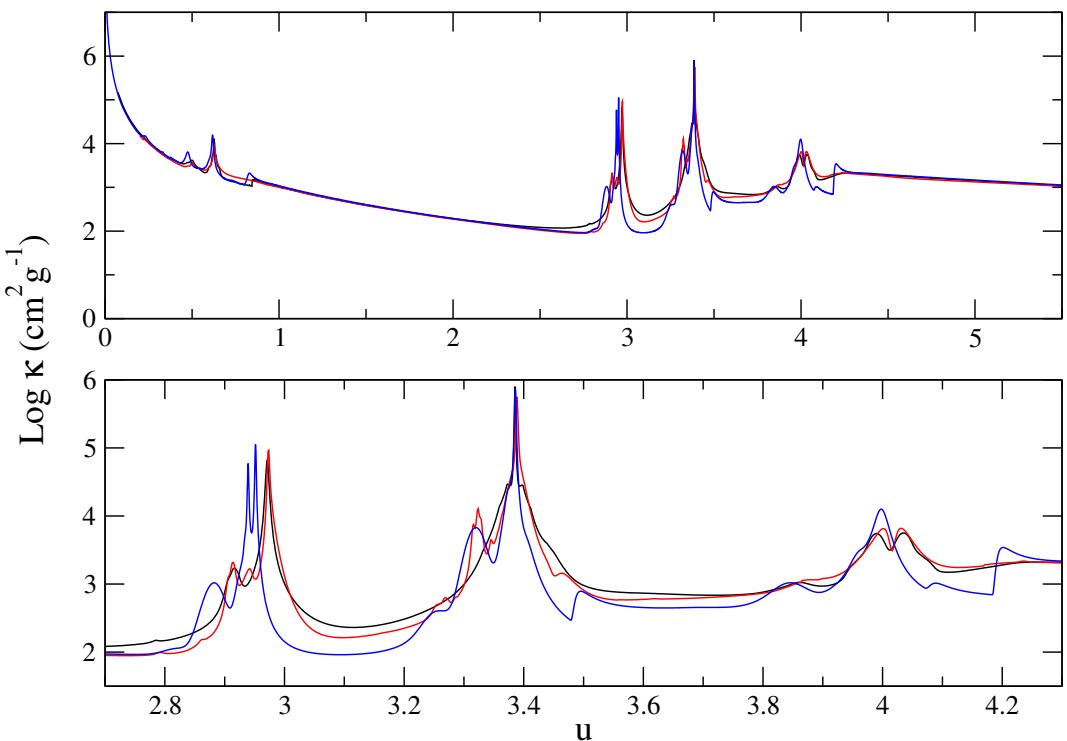

**Figure 10.** Oxygen monochromatic opacities (corrected for stimulated emission) as a function of the reduced photon energy $u = h\nu/kT$ in the ranges $0 \le u \le 5.5$ (**upper** panel) and $2.7 \le u \le 4.3$ (**lower** panel). Black curve: OP [4]. Red curve: OPLIB [97]. Blue curve: STAR [27].

As listed in Table 3, the dominant charge states are $O^{7+}$, $O^{8+}$, and, to a lesser extent, $O^{6+}$ and $O^{5+}$. Since the Rosseland weighting function peaks at $u = 3.83$, the line structure in the spectral interval $2.5 \le u \le 4.5$ of Figure 10 is of high relevance: O VII Kα, O VIII Lyα, and O VIII Lyβ at $u \approx 2.98$, 3.39, and 4.02, respectively. Interesting spectral features are the satellite-line arrays in the red wings of Kα (mainly from O VI Kα) and Lyα (mainly from O VI Kβ), the latter clearly missing from the OP curve probably due to the completeness issues discussed in Section 5.1.3. By comparing the OP and OPLIB curves in detail (see Figure 10 lower panel), we find that the FWHM of the Kα, Lyα, and Lyβ lines are similar, but the OP lines have more extended wings; in contrast, the STAR corresponding

lines are distinctively narrower. Therefore, at least for oxygen the discrepant broadening of the OP K lines seems to be due more to the red-wing corrections discussed in [98–100] and amply applied to OP monochromatic opacities than to narrower Stark widths, making the oxygen RMO in these plasma conditions somewhat larger ($\sim$15%) than those of STAR and OPLIB (see Table 3).

**Table 3.** RMO and ionization fractions for a pure oxygen plasma at $T = 192.91$ eV and $N_e = 10^{23}$ cm$^{-3}$. OP: [4]. STAR: [27]. OPLIB: [97].

| Method | RMO (cm$^2$ g$^{-1}$) | Ionization Fraction (O$^{(8-n)+}$) | | | | |
|--------|--------|--------|--------|--------|--------|--------|
| | | $n = 0$ | $n = 1$ | $n = 2$ | $n = 3$ | $n = 4$ |
| OP | 423 | 0.415 | 0.471 | $1.09 \times 10^{-1}$ | $5.05 \times 10^{-3}$ | $1.52 \times 10^{-4}$ |
| STAR | 357 | 0.423 | 0.447 | $1.16 \times 10^{-1}$ | $1.30 \times 10^{-2}$ | $8.09 \times 10^{-4}$ |
| OPLIB | 374 | 0.446 | 0.451 | $9.59 \times 10^{-2}$ | $6.23 \times 10^{-3}$ | $1.68 \times 10^{-4}$ |

## 6. Concluding Remarks

In the present report we have reviewed the computational methods used in earlier opacity revisions of the 1980–1990s to produce fairly reliable Rosseland-mean and radiative acceleration tables that have been used to model satisfactorily a variety of astronomical entities, among them the solar interior and pulsating stars. However, more recent developments, e.g., the revision of the solar photospheric abundances and the powerful techniques of helio and asteroseismolgy, have led to a serious questioning of their accuracy.

This critical crossroads has induced extensive revisions of the opacity tables and numerical frameworks, the introduction of novel computational methods (e.g., the STA method) to replace the traditional detailed-line-accounting approaches, and laboratory experiments that simulate the plasma environment of the solar CZB. They have certainly deepened our understanding of the contributing absorption processes but have not yet resulted in definite missing opacities; in this respect, alternative experimental attempts to reproduce the larger-than-expected opacity measurements of [33] are currently in progress [101,102].

Essential considerations of astrophysical opacity computations are accuracy and completeness in the treatment of the radiative absorption processes, both difficult to accomplish in spite of the powerful computational facilities available nowadays. As reviewed in Section 4, it has been shown that configuration interaction (CI) effects are noticeable in the spectrum shape in certain thermodynamic regimes for the topical cases of Cr, Fe, and Ni [14,17,19], while incomplete configuration accounting leads to sizable discrepancies in others [31]. To rigorously satisfy both requirements with the OP *R*-matrix method for isoelectronic sequences with electron number $N > 13$, it implies close-coupling expansions that soon become computationally intractable and must then be tackled with simpler CI methods (e.g., AUTOSTRUCTURE) that neglect the bound—continuum coupling. Taking into account the original satisfactory agreement between OPAL and OP as discussed in Section 2.3, perturbative methods such as the former appear to have an advantage in opacity calculations insofar as being able to manage configuration accounting more exhaustively. The introduction of powerful numerical frameworks (e.g., STA) to replace detailed line accounting reinforces this assertion. On the other hand, as discussed in Section 2.4, the OP *R*-matrix approach generates as a byproduct an atomic radiative database (e.g., TOPbase) of sufficient accuracy and completeness to be useful in a wide variety of astrophysical problems.

An inherent limitation of the OP *R*-matrix method in opacity calculations is that radiative properties are calculated assuming isolated atomic targets, i.e., exempt from plasma correlation effects that, as previously pointed out [103–106], could significantly modify the ionization potentials, excitation energies, and the bound–free and free–free absorption cross sections. In this respect, it has been recently shown [107] that ion–ion plasma correlations lead to $\sim$10% and $\sim$15% increases of the RMO in the solar CZB and in a pure iron plasma, respectively, that could be on their way to solve the

solar abundance problem. Plasma interactions by means of Debye–Hückel and ion-sphere potentials have already been included in atomic structure CI codes such as GRASP92 and AUTOSTRUCTURE, and are currently being used to estimate the plasma effects on the atomic parameters associated with Fe K-vacancy states [108].

**Acknowledgments:** I would like to thank Carlos Iglesias (Lawrence Livermore National Laboratory), James Colgan (Los Alamos National Laboratory), and Manuel Bautista (Western Michigan University) for reading the manuscript and for giving many useful suggestions. I am also indebted to James Colgan and Menahem Krief (Racah Institute of Physics, the Hebrew University of Jerusalem) for readily sharing their oxygen monochromatic opacity tabulations and ionization fractions. Private communications with Anil K. Pradhan (Ohio State University) and Sunny Vagnozzi (Stockholm University) are acknowledged. This work has been in part supported by NASA grant 12-APRA12-0070 through the Astrophysics Research and Analysis Program.

**Conflicts of Interest:** The author declares no conflict of interest.

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
