# Peer review of "Computation of Atomic Astrophysical Opacities"

_atoms, doi:10.3390/atoms6020028_

Round 1

Reviewer 1 Report

The MS is a comprehensive and up-to-date Review Article of the state-of-play in the highly topical area of opacities, mostly for astrophysics, which is the subject of much current debate. The Review does well to cover pretty much all aspects. The only omission that I detected was lack of reference to the work of Jeffrey & Saio 2006 MNRAS v371, 659 and v372 L48 whose study of instabilities tended to favor OP over OPAL.

Author Response

Comment: "The only omission that I detected was lack of reference to the work of Jeffrey & Saio 2006 MNRAS v371, 659 and v372 L48 whose study of instabilities tended to favor OP over OPAL."

Action: The sentence "In this respect, the temperature at which the Fe and Ni contributions to the Z-bump occur and their magnitudes have been shown to be critical in p- and g-mode pulsations in cool subdwarf B stars [72,73]." was included in the second paragraph of Section 4. 

72. Jeffery, C.S.; Saio, H. Fe-bump instability: the excitation of pulsations in subdwarf B and other low-mass stars. Mon. Not. R. Astron. Soc. 2006, 371, 659–672.

73. Jeffery, C.S.; Saio, H. Gravity-mode pulsations in subdwarf B stars: a critical test of stellar opacity. Mon. Not. R. Astron. Soc. 2006, 372, L48–L52.

Reviewer 2 Report

The manuscript “Computation of Atomic Astrophysical Opacities” includes a review of the differences between OP and OPAL opacity calculation methods. This paper, to my opinion, is called for due to the “Solar Composition problem”, and since recent studies have shown that element by element differences in opacity calculations can reach 25%. The paper is generally written well and clear and I recommend publication. 

However, there are a few comments:

A more explicit description is needed of the element by element comparison of opacities (see Blancard et al, Krief et al), since this shows much worse agreement than the total opacity of the solar composition. .

In section 2, a short explanation is needed regarding HOW does one use and need EOS and atomic data in calculating opacities. 

The sentence in row 221 is unclear: " However, the line-broadening dependency of the solar

222  opacity near the bottom of the convection zone has been previously reported only as moderate, i.e. to

223  within a few percent [51 ,84 ], and thus this new finding only seems to confirm it.”

In many places during the paper, the author write about comparisons between codes at temperature of 10-30 eV. It is unclear to me how are these relevant and why should one look at this regime when considering the Sun. It is different qualitatively due to different relative importance of different atomic processes (bound-bound, bound-free, free-free, scattering), as well as considerably different correlation effects contribution and complications. This is increased also due to the fact that the physics in both OP and OPAL is modeled parametrically. 

In general, Section 5 is read as “expert only”. This is a very important subject, that should be explained better. In particular it is important to explain what are the specific physics contributions, and how are they affecting line shapes in particular, and opacity calculatio. 

Since the main difference between OP and other calculation methods are in the line broadening, a comparison of the detailed spectrum of OP with other calculations in this respect is needed.

Author Response

Comment: A more explicit description is needed of the element by element comparison of opacities (see Blancard et al, Krief et al), since this shows much worse agreement than the total opacity of the solar composition.

Action: the last sentence of the second paragraph of Section 4 (page 8) now reads, “Problems with both the OPAL and OP tables have been reported; for instance, there are considerable differences in the OPAS and OP single-element monochromatic opacities although OPAS Rosseland mean opacities for the solar mixture by [59] agree to within 5% with both OPAL and OP for the entire Sun’s radiative zone (0.0 r/R0.713) [13]. The larger κ^OPAS_R /κ^OP_R ratios are found in Mg (44%), Fe (+40%), and Ni (+47%) due to variations in the ionic fractions (particularly for the lower charge states), missing configurations in OP, and the pressure broadening of the Kα line in the He-like systems.”

Comment: In section 2, a short explanation is needed regarding HOW does one use and need EOS and atomic data in calculating opacities. 

Action: A whole introductory segment has been included in Section 2 to explain the need for massive atomic data sets, an equation of state and line broadening.

Comment: The sentence in row 221 is unclear: " However, the line-broadening dependency of the solar opacity near the bottom of the convection zone has been previously reported only as moderate, i.e. towithin a few percent [51 ,84], and thus this new finding only seems to confirm it.”

Action: The paragraph “The line broadening approximations implemented in most opacity calculations have been recently questioned [27], reporting large discrepancies between the OP K-line widths and those in other opacity codes. It is also shown therein that the solar opacity profile is sensitive to the pressure broadening of K lines that can be empirically matched with the helioseismic indicators by a K-line width increase of around a factor of 100. However, the line-broadening dependency of the solar opacity near the bottom of the convection zone has been previously reported only as moderate, i.e. to within a few percent [51,86], and thus this new finding only seems to confirm it.” has been replaced by “The line broadening approximations implemented in most opacity calculations have been recently questioned [27], reporting large discrepancies between the OP K-line widths and those in other opacity codes. It is also shown therein that the solar opacity profile is sensitive to the pressure broadening of K lines, which can be empirically matched with the helioseismic indicators by a K-line width increase of around a factor of 100. This moderate line-broadening dependency (a few percent) of the solar opacity near the bottom of the convection zone concurs with previous findings [51,86].”

Comment: In many places during the paper, the author write about comparisons between codes at temperature of 10-30 eV. It is unclear to me how are these relevant and why should one look at this regime when considering the Sun. It is different qualitatively due to different relative importance of different atomic processes (bound-bound, bound-free, free-free, scattering), as well as considerably different correlation effects contribution and complications. This is increased also due to the fact that the physics in both OP and OPAL is modeled parametrically. 

Action: It is worth mentioning that this review is not only dedicated to solar opacities but more on atomic opacities in general. The comparisons at the lower temperatures of 10-30 eV were actually carried out to denote, for instance, the importance of 3-3 transitions in Fe third-row ions [46]. Other comparisons at low temperatures are invoked to estimate the importance of configuration effects. So, I would like to keep them in the report.

Comment: In general, Section 5 is read as “expert only”. This is a very important subject, that should be explained better. In particular it is important to explain what are the specific physics contributions, and how are they affecting line shapes in particular, and opacity calculatio. 

Action: If I may, I would like to keep the text on spectator-electron processes (old Section 5 and now Section 5.1) as is.

Comment: Since the main difference between OP and other calculation methods are in the line broadening, a comparison of the detailed spectrum of OP with other calculations in this respect is needed.

Action: Section 5 has been restructured to include, as suggested, a new section on line broadening in oxygen monochromatic opacities.

Reviewer 3 Report

The manuscript presents a readable and comprehensive overview of recent developments in opacity calculations, and their relevance to a variety of astrophysical phenomena. I have only a few optional suggestions for clarification and expansion.

In Section 4, the reader may be interested in the nature of the "puzzling side effects" and "[substantial modifications]" to opacity profiles.

In Figures 7 and 8, it is unclear why damping/broadening of the photoabsorption cross sections increases the peak values from the undamped curves: this seems to violate the preservation of line strength under changes to line profiles. Do the undamped cross sections exclude some processes?

In the discussion of Fig. 9, it may be worth noting that a similar incompleteness in the target states is the heart of the "serious question[ing]" of recent R-Matrix calculations (Refs. 28-30, 88).

Author Response

Comment: In Section 4, the reader may be interested in the nature of the "puzzling side effects" and "[substantial modifications]" to opacity profiles.

Action: the last sentence of the fourth paragraph in Section 4, “… but the observed frequency ranges can only be modeled with substantially modified mean-opacity profiles that are nevertheless impaired by puzzling side effects [76].” has been replaced by “… but the observed frequency ranges can only be modeled with substantially modified mean-opacity profiles (an increase of a factor of 3 or larger at log(T)=5.47 to ensure g-mode instability and a reduction of 65% at log(T) = 5.06 to match the empirical f non-adiabatic parameters) that are nevertheless impaired by puzzling side effects (enhanced convection efficiency in the Z bump that affects mode instability; avoided-crossing effects in radial modes) [76].”

Comment: In Figures 7 and 8, it is unclear why damping/broadening of the photoabsorption cross sections increases the peak values from the undamped curves: this seems to violate the preservation of line strength under changes to line profiles. Do the undamped cross sections exclude some processes?

Action:at the end of the second paragraph of page 13, the following sentence was added, “(It must be mentioned that resonances in Figs.7-8 are not fully resolved, so in some cases their peak values may appear to be underestimated suggesting departures from line-strength preservation.)”

Comment:In the discussion of Fig. 9, it may be worth noting that a similar incompleteness in the target states is the heart of the "serious question[ing]" of recent R-Matrix calculations (Refs. 28-30, 88).

Action: An additional paragraph was included after the first paragraph of page 15, “These difficulties of the R-matrix method in obtaining adequate target representations in spectator-electron processes, particularly when involving excited states, are at the core of the controversy around the extended R-matrix calculation of Fe XVII L-shell photoabsorption [2830,90] mentioned in Section 4.1.